# Partial annotation-based organs and tumor segmentation with progressive weakly supervised learning

Dengqiang Jia[1][0000−0002−0902−1882] and Zilong Wang[2]

[1] Hong Kong Centre for Cerebro-cardiovascular Health Engineering (COCHE), HongKong, China
[2] School of Electronic Information and Electrical Engineering, Shanghai Jiao Tong University, Shanghai, China
{dqjia}@hkcoche.org

**Abstract.** In medical image analysis, obtaining labeled data is expensive and time-consuming. Numerous unlabeled data can be used for efficient abdominal organ segmentation. Besides, partially annotated data is easier to collect and can be used to develop label-efficient algorithms, reducing the annotation cost for considerable performance. We proposed a progressive weakly supervised learning for abdomen organs and tumor segmentation, i.e., PWS-Seg. PWS-seg can learn from organs to tumors via a progressive framework based on partially annotated images. Moreover, we applied a class-wise label fusion strategy to get a new set of reliable pseudo labels. On the FLARE2023 online validation cases, with the help of unlabeled data, our method obtained the average dice similarity coefficient (DSC) of 82.68% and average normalized surface distance (NSD) of 86.00%, which is better than the method only using partial annotated. The average running time is 100.17s per case in the inference phase, and the maximum used GPU memory is 4128 MB.

**Keywords:** Partial label · Self-training · Organ segmentation.

## 1 Introduction

Supervised segmentations in medical image analysis depend on the quantity and quality of manual voxel-level labels, which is time-consuming and expensive because of the requirement of professional domain knowledge.

In more common scenarios where there is a small amount of labeled data and a large amount of unlabeled data [5, 6], semi-supervised segmentation is effective [2, 3, 23]. Larger medical image datasets are provided for researchers, e.g., FLARE22 Challenge [18], to develop the semi-supervised segmentation.

Besides the semi-supervised segmentation, label-effective image segmentation has attracted attention due to the relaxation of the need for dense labels to weak or partial labels, resulting in the weakly supervised segmentation (WSS) [16,17, 26]. The bounding boxes, scribbles, and partial labels are the most commonly used supervision types for WSS [26]. Among all these types, partial annotations,

i.e., only one or a few classes are labeled in images, can provide the flexibility to allocate the workload and have the potential to reduce the annotation cost.

However, it is difficult to directly learn from partially labeled datasets via traditional fully supervised learning frameworks. To segment multiple organs, existing methods made efforts to learn from partially annotated datasets.

When training with partially-labeled images, Zhou et al., tried to incorporate anatomical priors from fully-labeled dataset [31]. Fang et al., proposed the target adaptive loss to learn a unified multi-organ segmentation model from partially- and fully-labeled datasets [7]. Huang et al., trained multiple binary segmentation models using partially-labeled images and then learned a multi-organ network using pseudo labels from the binary models [12]. Zhang et al., used a dynamic segmentation head and a task-specific controller to address the partial annotation issues [30].

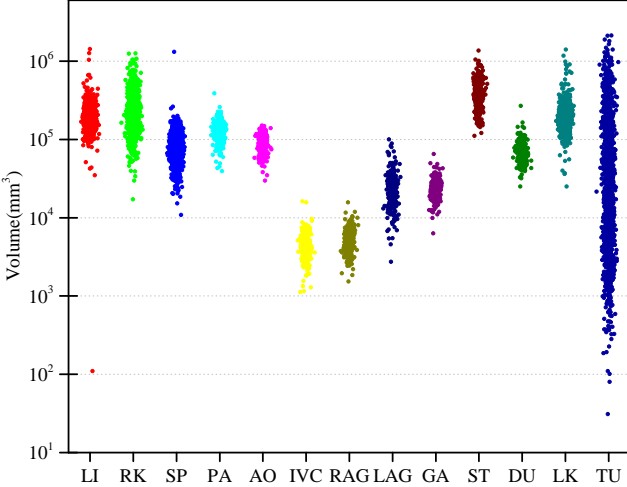

**Fig. 1.** Volume statistics of organs and tumors for partially labeled data of FLARE23. The labels in the horizontal coordinate are the abbreviations of the names of 13 organs (from Liver(LI) to Left Kidney(LK)) and tumor.

FLARE23 is different from FLARE22 in two ways; the first is that FLARE23 is partially labeled, and the second is that FLARE23 requires segmenting the tumor. One of the reasons why the latter is more difficult than organ segmentation is due to its uncertain location and state. For example, as shown in Figure 1, the volume distribution of the tumor (TU) has a larger variance than that of the other 13 organs.

In this paper, we propose a self-training strategy, i.e., a progressive weakly supervised learning framework (PWS-Seg), to segment the abdomen organs and tumors based on partial label annotation. The framework consists of two

stages, i.e., the organ segmentation (OS) stage and the organ-tumor segmentation (OTS) stage. The OS stage extracts the features of multiple organs from the limited labeled organs and then propagates the knowledge to unlabeled regions. The OTS stage uses the consistency of organ prediction during the progressive self-training process to correct the pseudo labels of organs and tumors.

## 2   Method

Figure 2 shows the diagram of our two-stage approach.

We propose a progressive weakly-supervised segmentation (PWS-Seg) framework for multi-organ and tumor segmentation tasks, which can leverage partial annotated data and numbers of unlabeled data. Since we assume that the abdominal tumors were highly spatially related to the abdominal organs, we introduce a two-stage segmentation to complete the organ-tumor segmentation task. The two networks have similar UNet-based architectures.

Algorithm 1 presents the scheme for PWS-Seg. In the first stage, we train the organ segmentation network (OS CNN) twice (Section 2.3). We can use each of the trained OS CNNs to predict the pseudo labels of organs of unlabeled data and partially annotated data. OS CNN was trained using our previous work, i.e., a cross-supervision method [15].

In the second stage, we also train the organ and tumor segmentation network (OTS CNN) twice. We can use each of the trained CNNs to predict the pseudo labels of organs and tumors of unlabeled data and partially annotated data. To generate more reliable pseudo labels, we propose a fusion strategy for the pseudo labels and partially annotated data (Section 2.4).

### 2.1   Preprocessing

The annotated images were cropped as patches using their corresponding labels, which avoided using numerous patches without any labels. For the unlabeled images, we used the results of the OS stage to crop the abdominal organ regions. All the images were re-sampled for a fixed spacing, i.e., $2.5\text{mm} \times 0.8\text{mm} \times 0.8\text{mm}$.

### 2.2   Notation of partial label segmentation

Let $\mathcal{L} = \left\{ (\boldsymbol{x}^{(1)}, \boldsymbol{y}^{(1)}), (\boldsymbol{x}^{(2)}, \boldsymbol{y}^{(2)}), ..., (\boldsymbol{x}^{(N)}, \boldsymbol{y}^{(N)}) \right\}$ and $\mathcal{U} = \left\{ \boldsymbol{x}^{(N+1)}, ..., \boldsymbol{x}^{(M)} \right\}$ denote the labeled data and unlabeled data. Here, we denote $\boldsymbol{x}$ as the intensity image, and denote $\boldsymbol{y}^{(i)}$ as annotation(label) image.

We can define the partial label data $\mathcal{P}$ based on $\mathcal{L}$. For a given element $(\boldsymbol{x}^{(i)}, \boldsymbol{y}^{(i)})$ in $\mathcal{L}$ and class $k$, we can define the partial annotated label image:

$$\boldsymbol{y}^{(i)} = [y_1^{(i)}, y_2^{(i)}, ..., y_K^{(i)}] \subseteq \{0, 1\}^K. \tag{1}$$

Here $y^{(i)} = 1$ and $y^{(i)} = 0$ mean the annotation of class $k$ is present and absent for $\boldsymbol{x}^{(i)}$, respectively. Based on the definitions of $\mathcal{P}$ and $\mathcal{L}$, the partially labeled

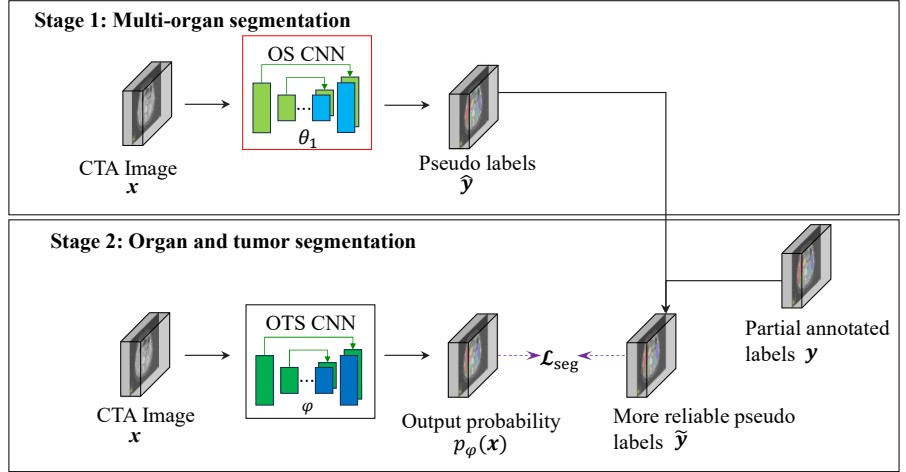

**Fig. 2.** The diagram of the two-stage organ and tumor segmentation framework. Stage 1 (OS stage): A fully annotated organ dataset and an unlabeled dataset are used to segment the organ using the cross-supervision method. The trained OS CNN ($p_{\theta_1}$) is used to predict pseudo labels of organs of the unlabeled images. The pseudo labels of organs, which can also seen as a set of partially annotated labels of organs and tumors, are combined with the given partially annotated labels to generate a set of more reliable pseudo labels. Stage 2 (OTS stage): Based on the more reliable dataset, we can segment the organs and tumors, simultaneously.

---

**Algorithm 1:** Framework of PWS-Seg.

---

**Input:** Partially annotated data: $\mathcal{P} = \{\mathcal{P}_O, \mathcal{P}_{OT}\}$; Unlabeled data:$\mathcal{U}$;
       Organ segmentation network:OS CNN($\theta$),Organ-tumor
       segmentation network:OTS CNN($\varphi$);

**Output:** $\varphi$.

Initialize $\theta$ and $\varphi$ according to Tables 2 and 3;

**foreach** *epoch in max-num-epoch* **do**
    | Sample batch from $\mathcal{P}_O$, $\mathcal{P}_{OT}$ and $\mathcal{U}$; $\mathcal{P}_{OT}$ in this step were
    |  considered as unlabeled data;
    | Train parameter $\theta$ of OS CNN;

Predict pseudo labels of organs for $\mathcal{U}$ and $\mathcal{P}_{OT}$, divide the tumor label
 into different organs;

**foreach** *epoch in max-num-epoch* **do**
    | Sample batch from $\mathcal{P}_O$, and sample batch from $\mathcal{P}_{OT}$ and $\mathcal{U}$ with
    |  their pseudo labels of organs;
    | Train parameter $\theta$ of OS CNN;

Predict pseudo labels of organs ($\hat{\mathcal{y}}$) for $\mathcal{P}_{OT}$ using trained OS CNN;
For $\mathcal{P}_{OT}$, abandon the data whose partial label did not contain tumor
 class and fuse the pseudo organ labels of organs and partial
 annotations, resulting in a set of data with more reliable pseudo labels
 $\tilde{\mathcal{y}}$;

**foreach** *epoch in max-num-epoch* **do**
    | Sample batch from $\tilde{\mathcal{y}}$; Train parameter $\varphi$ of OST CNN;

Predict pseudo labels of organs and tumors ($\hat{\mathcal{y}}$) for $\mathcal{P}_O$, $\mathcal{P}_{OT}$ and $\mathcal{U}$;
Fuse the partial labels and pseudo labels into $\tilde{\mathcal{y}}$;

**foreach** *epoch in max-num-epoch* **do**
    | Sample batch from $\tilde{\mathcal{y}}$ ;
    | Train parameter $\varphi$ of OST CNN;

---

data $\mathcal{P}$ is also the labeled data related to the target class $K$. In this paper, we assume that the tumors were unknown, such that those images where all the organs are fully labeled (i.e., $\mathcal{P}_O$) can be seen as partially annotated images. The images where organs and tumors are partially labeled are denoted as $\mathcal{P}_{OT}$.

The aim of the partial label segmentation is to obtain a segmentation plan that can leverage all the obtained data, i.e., $\mathcal{P}$ and $\mathcal{U}$. We can use the segmentation plan $\boldsymbol{P}$ to predict a probability map $p$ for $\boldsymbol{x}$ as:

$$\boldsymbol{P} = p_\varphi(\boldsymbol{x}). \tag{2}$$

### 2.3   Organ segmentation using cross supervision

In the organ and tumor segmentation task, fully annotated organ labels can also be seen as partial labels when we assume that the tumor class is absent.

In this stage, to segment abdominal organs with limited organ-labeled and un-labeled data $\mathcal{U}$, we use the cross-supervision method, which is presented in our previous work [15]. Two sub-networks ($p_{\theta_1}$ and $p_{\theta_2}$) are introduced. They have the same structures and the same number of parameters but are initialized differently at the beginning.

During the training procedure, to leverage the unlabeled data, the cross-supervised (CS) losses for organ-labeled and unlabeled data, i.e., $\mathcal{C}_u^u$ and $\mathcal{C}_u^l$, are introduced, respectively.

The training loss function for organ segmentation can be formulated as:

$$\mathcal{C} = \mathcal{C}_s + \mathcal{C}_u^u + \mathcal{C}_u^l. \tag{3}$$

In the first stage, we train the OS CNN twice. During the first training procedure, we input fully organ-labeled images ($\mathcal{P}_O$), partially labeled images($\mathcal{P}_{OT}$), and unlabelled images ($\mathcal{U}$). The partially labeled images are treated as unlabelled images in this stage. We use the first trained OS CNN to predict pseudo-labels for both partially labeled and unlabelled images. Since there are a certain number of correct organ labels in the partially labeled images, we retain the labeled organ labels and classify the tumor regions into organ categories when generating pseudo-labeled images for the partially labeled images. In other words, we only use data with organ labels in the second training of OS CNN. At the end of the OS stage, we use the second trained OS CNN to predict $\hat{\boldsymbol{y}}$ as shown in Figure 2. More details about this stage are provided in Algorithms 1.

### 2.4    Class-wise label fusion strategy for partially annotated labels

To obtain more reliable pseudo labels, we propose a class-wise label fusion strategy to use partially labeled data. Although cross-supervision strategy can leverage the unlabeled data $\mathcal{U}$, partially labeled data could not be used directly. We can generate more reliable pseudo labels of partially annotated images.

Given the partial labeled data $(\boldsymbol{x}^{(j)}, \boldsymbol{y}^{(j)})$ in $\mathcal{P}$ and its corresponding pseudo label is:

$$\hat{\boldsymbol{y}}^{(j)} = [\hat{y}_1^{(j)}, \hat{y}_2^{(j)}, ..., \hat{y_K}^{(j)}] \subseteq \{1\}^K. \tag{4}$$

Here, $\hat{\boldsymbol{y}}^{(j)} \subseteq \hat{\mathcal{Y}}$ can be generated by any teacher models. Based on partial label and pseudo label information, we can generate more reliable pseudo labels for partially labeled images: $\tilde{\boldsymbol{y}}^{(j)} = [\tilde{y}_1^{(j)}, \tilde{y}_2^{(j)}, ..., \tilde{y_K}^{(j)}] \subseteq \tilde{\mathcal{Y}}$:

$$\tilde{y_k}^{(j)} = \begin{cases} y_k^{(j)}, & \text{if } y_k^{(j)} = 1; \\ \hat{y}_k^{(j)}, & \text{others.} \end{cases} \tag{5}$$

In the second stage, we also train the OTS CNN twice. During the first training procedure, we use the label fusion strategy (Equation 5) to generate a set of more reliable pseudo labels for the partially annotated images. It should be noted that the partially annotated images, whose labels did not contain tumor class were abandoned, resulting in a set of data with more reliable pseudo labels

$\tilde{\mathcal{Y}}$. We use the first trained OTS CNN to predict pseudo-labels for both partially labeled and unlabelled images. Label fusion strategy and label selection strategy are also used to generate the second set of pseudo labels, which are used to train a second OTS CNN. More details about this stage are provided in Algorithms 1.

## 3 Experiments

### 3.1 Dataset and evaluation measures

The FLARE 2023 challenge is an extension of the FLARE 2021-2022 [20] [21], aiming to aim to promote the development of foundation models in abdominal disease analysis. The segmentation targets cover 13 organs and various abdominal lesions. The training dataset is curated from more than 30 medical centers under the license permission, including TCIA [4], LiTS [1], MSD [25], KiTS [10,11], autoPET [8,9], TotalSegmentator [27], and AbdomenCT-1K [22]. The training set includes 4000 abdomen CT scans where 2200 CT scans with partial labels and 1800 CT scans without labels. The validation and testing sets include 100 and 400 CT scans, respectively, which cover various abdominal cancer types, such as liver cancer, kidney cancer, pancreas cancer, colon cancer, gastric cancer, and so on. The organ annotation process used ITK-SNAP [28], nnU-Net [14], and MedSAM [19].

The evaluation metrics encompass two accuracy measures—Dice Similarity Coefficient (DSC) and Normalized Surface Dice (NSD)—alongside two efficiency measures—running time and area under the GPU memory-time curve. These metrics collectively contribute to the ranking computation. Furthermore, the running time and GPU memory consumption are considered within tolerances of 15 seconds and 4 GB, respectively.

### 3.2 Implementation details

**Environment settings** The development environments and requirements are presented in Table 1.

**Table 1.** Development environments and requirements.

| | |
|---|---|
| System | Ubuntu 20.04.4 LTS and Windows 11 |
| CPU | e.g., Intel(R) Core(TM) i9-13900K CPU@3.00GHz |
| RAM | 16×4GB; 2.67MT/s |
| GPU (number and type) | One NVIDIA 3090 24G |
| CUDA version | 11.0 |
| Programming language | Python 3.8 |
| Deep learning framework | torch 1.11.0 |

**Training protocols** We implemented the proposed framework using EfficientSeg-Net [29] and nnUnet [14] used in FLARE21 [20] and FLARE22 [18] challenge.

In the OS stage of our proposed method, we used 2250 partially labeled images (50 fully organ-annotated images from FLARE22 and 2200 partially labeled images from FLARE23) and 2800 (1000 from FLARE22 [18] and 1800 from FLARE23) unlabeled images. In this stage, we also employed a cascade strategy, which aimed to segment the abdomen organs via a coarse-to-fine procedure. Since the abdominal region is large, we can not efficiently segment all organs in a single-stage way. Therefore, we segmented the organs from down-sampled images in the first place, which can be seen as a coarse segmentation. With the help of coarse segmentation results, we segmented the organs of the original images.

In the OTS stage, we used 50 partially annotated images (fully organ-annotated images), 1496 partially annotated images (must have the tumor label), and 2800 (1000 from FLARE22 and 800 from FLARE23) unlabeled images to train the network.

We used the same processing strategy and data augmentation method for all images as in our previous work [15]. Crop, random rotation, random transition, and random elastic deformation were used for data augmentation. We randomly resampled the data with the size and spacing described in Table 2 and Table 3.

**Table 2.** Training protocols for OS stage.

| | |
|---|---|
| Network initialization | Kaiming normal initialization |
| Batch size | 8(coarse), 1(fine) |
| Input size (coarse) | 160×160×160 |
| Input size (fine) | 192×192×192 |
| Total epochs | 500(coarse), 1000(fine) |
| Optimizer | Adam with betas (0.9, 0.99), L2 penalty: 0.00001 |
| Loss | Dice loss and focal loss (alpha = 0.5, gamma = 2) |
| Initial learning rate (lr) | 0.01 |
| Training time (coarse) | 6 (coarse), 300(fine) hours |

## 4   Results

The results show that the method using unlabeled data improves the dice score of the method with partially labeled images.

Table 4 shows the results of the proposed methods on the validation dataset. The results of our submitted solution (docker container), which was evaluated by the organizers of FLARE2023, are reported in Table 4. The public validation dataset contains 50 cases, while the online validation dataset contains 100 cases.

**Table 3.** Training protocols for OTS stage.

| | |
|---|---|
| Network initialization | Kaiming normal initialization |
| Batch size | 2 |
| Patch size | 48×192×192 |
| Total epochs | 1000 |
| Optimizer | Adam |
| Initial learning rate (lr) | 0.01 |
| Training time | 672 hours |
| Loss function | Cross-entropy and Dice loss |
| Number of model parameters | 48.84M[3] |
| Number of flops | 995.46G[4] |

**Table 4.** Quantitative results PWS-Seg($\mathcal{P}+\mathcal{U}$) in terms of DSC and NSD on the validation dataset. There are 50 validation cases in the public validation and 100 cases in the online validation. We report the mean and standard deviation with ±.

| Target | Public Validation | | Online Validation | | Testing | |
|---|---|---|---|---|---|---|
| | DSC(%) | NSD(%) | DSC(%) | NSD(%) | DSC(%) | NSD (%) |
| Liver | 97.21 ± 1.36 | 96.36 ± 4.55 | 97.24 | 96.66 | | |
| Right Kidney | 94.47 ± 6.31 | 94.30 ± 8.02 | 94.07 | 94.12 | | |
| Spleen | 96.28 ± 3.03 | 97.14 ± 5.46 | 96.47 | 97.35 | | |
| Pancreas | 81.74 ± 5.26 | 93.38 ± 3.60 | 79.65 | 91.54 | | |
| Aorta | 95.69 ± 3.68 | 97.86 ± 4.70 | 95.97 | 98.09 | | |
| Inferior vena cava | 92.44 ± 4.10 | 94.23 ± 4.93 | 92.21 | 93.75 | | |
| Right adrenal gland | 80.35 ± 6.17 | 91.07 ± 5.29 | 79.73 | 91.10 | | |
| Left adrenal gland | 75.79 ± 10.49 | 84.46 ± 10.58 | 75.29 | 83.67 | | |
| Gallbladder | 77.61 ± 25.28 | 74.85 ± 25.43 | 78.05 | 74.59 | | |
| Esophagus | 79.41 ± 15.57 | 90.19 ± 15.06 | 80.21 | 91.24 | | |
| Stomach | 90.48 ± 6.98 | 92.52 ± 9.59 | 90.02 | 91.75 | | |
| Duodenum | 77.32 ± 9.20 | 91.67 ± 6.10 | 77.97 | 91.88 | | |
| Left kidney | 93.01 ± 10.26 | 91.85 ± 13.09 | 92.55 | 91.83 | | |
| Tumor | 34.42 ± 33.26 | 27.40 ± 27.47 | 28.05 | 16.40 | | |
| Average(Organ) | 87.06 ± 8.02 | 91.53 ± 5.85 | 86.88 | 91.35 | | |
| Average | 83.30 ± 15.61 | 86.95 ± 17.45 | 82.68 | 86.00 | | |

**Table 5.** Quantitative results of i.e., PWS-Seg in terms of DSC and NSD on the validation dataset. The symbol 1540 ($\mathcal{P}$) denotes the method using 50 fully annotated organ images $\mathcal{P}_O$ and 1496 partially annotated organ-tumor images $\mathcal{P}_{OT}$. PWS-Seg$^*$($\mathcal{P}$) denotes the method without using label fusion strategy. PWS-Seg$^\dagger$($\mathcal{P}+\mathcal{U}$) denotes the method with larger sized network. We report the mean and standard deviation with ±.

| Organ/Tumor | PWS-Seg$^*$($\mathcal{P}$)
1540($\mathcal{P}$)=50+1496
DSC(%), NSD(%) | PWS-Seg($\mathcal{P}$)
1540($\mathcal{P}$)=50+1496
DSC(%), NSD(%) | PWS-Seg$^\dagger$($\mathcal{P}+\mathcal{U}$)
1540($\mathcal{P}$)+2800($\mathcal{U}$)
DSC(%), NSD(%) |
|---|---|---|---|
| Liver | 96.52±2.21,95.50±6.04 | 96.86±3.29,96.49±4.20 | 97.71±0.62,97.81±1.94 |
| RK | 92.02±9.34,90.55±10.99 | 91.04±15.59,90.15±16.42 | 94.94±7.18,94.93±7.75 |
| Spleen | 94.79±7.42,94.32±8.00 | 95.77±4.54,96.30±6.63 | 96.36±3.45,97.11±5.44 |
| Pancreas | 78.28±6.73,89.72±5.67 | 82.08±6.11,93.71±4.87 | 83.20±4.91,94.68±3.22 |
| Aorta | 96.14±3.06,98.43±4.40 | 94.84±4.32,97.49±5.28 | 95.73±3.96,97.97±4.71 |
| IVC | 92.69±4.03,94.97±5.13 | 92.74±4.38,94.54±5.19 | 92.75±3.90,94.42±4.87 |
| RAG | 81.72±6.04,91.61±4.87 | 74.41±10.12,86.38±7.71 | 83.42±5.60,92.84±4.40 |
| LAG | 74.97±13.58,84.93±15.20 | 74.91±10.47,83.57±9.88 | 78.49±9.76,87.31±9.17 |
| Gallbladder | 77.33±25.78,76.26±25.64 | 78.19±24.62,77.54±26.19 | 83.27±20.08,81.23±21.11 |
| Esophagus | 76.85±15.38,89.65±13.74 | 77.32±16.26,87.40±15.92 | 80.04±15.07,90.75±14.23 |
| Stomach | 87.97±9.18,89.38±11.12 | 89.83±7.10,92.76±8.95 | 91.25±6.25,94.30±7.40 |
| Duodenum | 77.89±7.79,92.11±5.98 | 78.95±8.63,92.87±6.13 | 79.58±8.71,93.18±6.02 |
| LK | 87.41±17.53,83.35±18.39 | 91.57±12.15,91.44±13.83 | 93.33±12.19,93.08±14.20 |
| Tumor | 31.10±33.89,25.40±27.84 | 37.72±33.76,35.18±29.26 | 44.19±34.26,40.06±30.85 |
| Avg. | 81.83±15.97,60.45±35.05 | 82.59±14.72,86.84±15.29 | 85.30±13.21,89.26±14.29 |

## 4.1   Quantitative results on validation set

In Table 4, the average DSC of organ segmentation on public and online validation datasets are separately 87.06% and 86.88%, which demonstrates that our method shows robust performance on organ segmentation. However, the DSC of tumor segmentation on public and online validation datasets are separately 34.42% and 28.05%, which shows that tumor segmentation remains challenging.

Compared to the method of only using partially annotated data (PWS-Seg($\mathcal{P}$) ) shown in Table 5), using unlabeled data, the PWS-Seg improves the average DSC from 82.59% to 83.30%, while the average NSC improves from 86.84% to 86.95%, which is consistent with the conclusion of FLARE22 challenge [18].

Table 4 shows that the Tumor, LAG, Duodenum, and Gallbladder are the three difficult regions, while the Liver, Spleen, and Aorta are the three easy organs for abdominal organ and tumor segmentation. The difficulties may be due to unclear boundaries, class imbalanced issues, and large variations of shapes. Besides, Table 4 and Table 5 show that the standard deviations of the tumor and Gallbladder segmentation are relatively large, which demonstrates the method achieves disappointed robustness for Tumor and Gallbladder.

To validate the effect of the label fusion strategy, we perform PWS-Seg with simple pseudo labels without any fusion, i.e., PWS-Seg*($\mathcal{P}$) in Table 5. The average DSC decreases from 82.59 % to 81.83%, which shows the efficacy of the label fusion strategy.

To validate the effect of the larger size of the network on the results, we increase the number of features to 32 and the patch size to $48 \times 224 \times 224$, denoted as PWS-Seg$^{\dagger}$($\mathcal{P}+\mathcal{U}$). The results in Table 5 show that a larger sized network can improve the accuracy of the segmentation, which is consistent with the conclusion in the work [13] of FLARE22.

As shown in Figure 3, in Case #0093, the segmentation results of our method have large boundary variations of Duodenum. Moreover, both in Case #0093 and Case #0033, our method fails to recognize the tumor of RK even if the tumor boundaries are clear.

**Table 6.** Quantitative evaluation of segmentation efficiency in terms of the running them and GPU memory consumption. Total GPU denotes the area under the GPU Memory-Time curve. Evaluation GPU platform: NVIDIA QUADRO RTX5000 (16G).

| Case ID | Image Size | Running Time (s) | Max GPU (MB) | Total GPU (MB) |
|---------|------------|------------------|--------------|----------------|
| 0001 | (512, 512, 55) | 65.13 | 3442 | 156868 |
| 0051 | (512, 512, 100) | 101.57 | 4402 | 358826 |
| 0017 | (512, 512, 150) | 117.42 | 4618 | 432704 |
| 0019 | (512, 512, 215) | 87.14 | 3874 | 262002 |
| 0099 | (512, 512, 334) | 122.24 | 4388 | 396838 |
| 0063 | (512, 512, 448) | 146.61 | 4602 | 489157 |
| 0048 | (512, 512, 499) | 150.43 | 4524 | 509211 |
| 0029 | (512, 512, 554) | 180.96 | 5196 | 692933 |
| Avg. (20 cases) | - | 100.17 | 4128 | 339753 |

### 4.2 Qualitative results on validation set

Figure 3 shows two examples with good segmentation results(#0038_#slice172 and #0053_#slice72) and two examples with bad segmentation results(#0093_#slice58 and #0033_#slice74) in the validation set.

### 4.3 Segmentation efficiency results on validation set

Table 6 presents the segmentation efficiency results of 8 cases, whose image sizes are increasing. The runtime of the case with the smallest image size, i.e., Case #0001, is 65.13 s. By contrast, the runtime of the case with the largest image size, i.e., Case #0029, is 180.96 s.

The mean runtime is 100.17 s per case in the prediction step, the maximum used GPU memory is 4128 MB, and the AUC GPU time is 339753 MB.

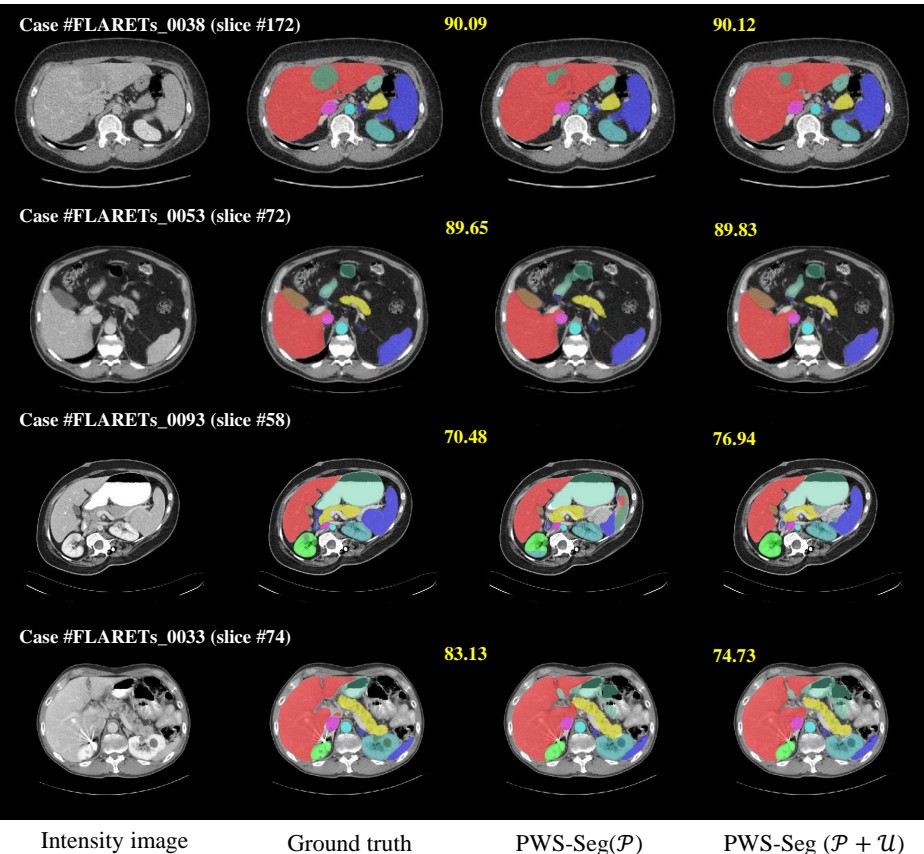

**Fig. 3.** Qualitative results on good (Case #0038 and Case #0053) and bad (Case #0093 and Case #0033) examples. The first column is the intensity image, the second column is the ground truth, and the third and the fourth columns are the results achieved by our proposed method. PWS-Seg($\mathcal{P}$) denotes the proposed method using only the partially annotated dataset. PWS-Seg($\mathcal{P}+\mathcal{U}$) denotes our proposed solution. The DSC of each case is presented in the top-left corner.

### 4.4    Results on final testing set

## 5    Discussion and conclusion

Using unlabeled data, the proposed progressive weakly supervised method achieved better results than the results of the method using the partially annotated data. Whichever method is used, the segmentation of some organs and tumors is still challenging. Tumors are highly variable in shape and appearance due to uncertainty of location and status. As shown in Figure 1, the volumes of tumors have larger variations than organs. Thus, tumor segmentation obtained disappointing performance because of uncertainties of locations, regularity of unremarkable shapes, unclear boundaries, number of individuals, etc. The existence of tumors in organs, such as Livers and Kidneys, are critical factor for poor organ segmentation performance. Besides, further research is needed to identify and use the remarkable image properties and shape patterns.

The proposed PWS-Seg model used over 1000 unlabelled images, but the performance of the method is limited by the amount of time-consuming training of the model using images of the same type. Future attention may need to be paid to how representative training samples can be filtered out of thousands of data.

### 5.1    Limitation and future work

We summarize the limitations and future work as follows:

– Efficiently extract the features of tumors and organs with large shape and appearance variations.
– Robust network trained with partially annotated labels.
– High-quality datasets which have enough diversity and common features.

**Acknowledgements** The authors of this paper declare that the segmentation method they implemented for participation in the FLARE 2023 challenge has not used any pre-trained models nor additional datasets other than those provided by the organizers. The proposed solution is fully automatic without any manual intervention. We thank all the data owners for making the CT scans publicly available and CodaLab [24] for hosting the challenge platform.

This work was supported fully by InnoHK Project at Hong Kong Centre for Cerebro-cardiovascular Health Engineering (COCHE).

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

**Table 7.** Checklist Table. Please fill out this checklist table in the answer column.

| Requirements | Answer |
| --- | --- |
| A meaningful title | Yes/No |
| The number of authors ($\leq 6$) | Number |
| Author affiliations and ORCID | Yes/No |
| Corresponding author email is presented | Yes/No |
| Validation scores are presented in the abstract | Yes/No |
| Introduction includes at least three parts: background, related work, and motivation | Yes/No |
| A pipeline/network figure is provided | Figure number |
| Pre-processing | Page number |
| Strategies to use the partial label | Page number |
| Strategies to use the unlabeled images. | Page number |
| Strategies to improve model inference | Page number |
| Post-processing | Page number |
| Dataset and evaluation metric section is presented | Page number |
| Environment setting table is provided | Table number |
| Training protocol table is provided | Table number |
| Ablation study | Page number |
| Efficiency evaluation results are provided | Table number |
| Visualized segmentation example is provided | Figure number |
| Limitation and future work are presented | Yes/No |
| Reference format is consistent. | Yes/No |