# OpenReview forum: "Partial annotation-based organs and tumor segmentation with progressive weakly supervised learning"
_MICCAI.org/2023/FLARE — Submitted to FLARE 2023_

### Official Review · Reviewer_nwAU · 2023-09-21
**Partial annotation-based organs and tumor segmentation with progressive weakly supervised learning**

**Rating:** 7
**Confidence:** 5

**Review:**

Summary

This paper presents a progressive partial label learning framework that includes cross-supervision and a strategy for class-wise label fusion with pseudo label.
The results indicate that the method achieves a high level of accuracy for 13 abdominal organs.

Comments
1. Typographical errors, e.g. Fig. 2 occupies a full page, Table 4 and Table 5 are out of bounds. Table 7 needs to be completed.
2. Spelling error. NSD should replace NSC, "while the average NSC improves" in the experimental section. And U-Net should be used instead of UNet.
3. The methods section does not describe post-processing and methods to improve the efficiency of inference.

---

> ### Comment · Reviewer_nwAU · 2023-11-20
> **2nd round (final) Review**
>
> Currently, the entire article is incomplete.
>
> The authors have not revised the paper to respond to the comments of the reviewers. The authors should submit a new version of the paper.

---

### Official Review · Reviewer_Q8J7 · 2023-09-26

**Rating:** 4
**Confidence:** 5

**Review:**

This paper presents a compelling analysis to point out the difficulty of the FLARE23 dataset. The methodology is demonstrated clearly and experiments are sufficient, offering a well-organized paper that provides a solution for segmentation tasks with weakly labeled data.

However, my major concern is that **the proposed method used an additional FLARE22 dataset**, especially for using the labeled data part. It's unfair to compare other works in this competition.  But to take a step back, if the author can give experimental results to support that the method is still efficacious (consistent performance or slightly worse than this version) without using the FLARE22 labeled data, this is still a highly recommended paper.

---

### Official Review · Reviewer_29MC · 2023-09-27
**Review for "Partial annotation-based organs and tumor segmentation with progressive weakly supervised learning"**

**Rating:** 6
**Confidence:** 5

**Review:**

This is a well-written paper that presents a progressive approach to achieving good segmentation results on partially annotated datasets. However, I have several suggestions:

The paper should clearly indicate the corresponding author, even if there are only two authors.

The paper lacks a description of how inference acceleration and memory usage reduction were achieved.

Some tables exceed the width of the page and need to be adjusted.

In Figure 3, the spacing between each image should be consistent.

---

### Official Review · Reviewer_6Lmm · 2023-09-29
**Review for "Partial annotation-based organs and tumor segmentation with progressive weakly supervised learning"**

**Rating:** 4
**Confidence:** 4

**Review:**

The paper is well-organized and presents progressive cross-supervision and class-wise pseudo label fusion strategy, which highlight the weakly-supervised learning framework. The experiments are sufficient, and the method achieves a high-level accuracy.

However, there are some concerned issues in this paper.
In the Algorithm, after first train of OS CNN predict pseudo labels of organs for U and Pot, divide the tumor label into different organs. Do you sort out tumors to organs? How do you label them? How does this step influence the final segmentation? This should be elucidated.
Another concern is that use of additional FLARE22 datasets. In my opinion, authors should demonstrate the efficiency of their method without using the FLARE22 data, especially without labelled.

---

### Official Review · Reviewer_DRR9 · 2023-10-03
**Review for "Partial annotation-based organs and tumor segmentation with progressive weakly supervised learning"**

**Rating:** 5
**Confidence:** 5

**Review:**

The paper proposed a progressive weakly supervised learning for abdomen organs and tumor segmentation with supervision of partially labelled dataset and demonstrated that using unlabeled data would result in better results. But there are also some issues that need to be noted.
1. In terms of completeness, the article did not mention their strategies to improve inference speed and reduce resource consumption.
2. Table 4 and 5 are visually unattractive and go beyond the layout of the template.

---

> ### Comment · Reviewer_DRR9 · 2023-11-30
> **2nd round Review**
>
> In terms of completeness, there are still significant issues with this article at present.
>
> The author did not respond to the previous review comments or make any modifications to their paper.
>
> Most importantly, the test metrics are not given in tabel 4.

---

### Official Review · Reviewer_Ha8f · 2023-10-04
**Review for "Partial annotation-based organs and tumor segmentation with progressive weakly supervised learning"**

**Rating:** 6
**Confidence:** 4

**Review:**

The paper propose a progressive weakly-supervised segmentation (PWS-Seg) frame-work for multi-organ and tumor segmentation tasks, which can leverage partial annotated data and numbers of unlabeled data. The paper is a nice work with complete structure. There are several minor issues:
1. The proposed method used an additional FLARE22 dataset, Challenge organizers don't seem to allow it.
2. Table 4 and Table 5 are too wide and out of the margins.
3. How to achieve inference acceleration and memory usage reduction, which is not mentioned in the paper

---

### Official Review · Reviewer_3k5o · 2023-10-23
**A progressively supervised segmentation method.**

**Rating:** 6
**Confidence:** 4

**Review:**

The methodology presented in this paper completes the organ tumor segmentation in each case in compliance with the official requirements. However, the following problems exist:
1. the paper uses additional data from FLARE2022, but it seems that the official does not allow to use it, it is recommended to discuss further with the official.
2. the layout of the paper has a big problem, some pictures and tables are beyond the margins, make sure to adjust them when you revise it
3. The loss function used is not reflected by a specific formula in the paper.

---

### Decision · Program_Chairs · 2023-10-24

**Decision:**

Reject

**Comment:**

The authors didn't make responses to the valuable review comments.